# DRIS Norms and Sufficiency Ranges for Persimmon 'Rojo Brillante' Grown under Mediterranean Conditions in Spain

Julia Morales [ID], Isabel Rodríguez-Carretero, Belen Martínez-Alcántara [ID], Rodolfo Canet [ID] and Ana Quiñones *[ID]

Centro Para El Desarrollo de la Agricultura Sostenible, Instituto Valenciano de Investigaciones Agrarias, 46113 Valencia, Spain; morales_jul@gva.es (J.M.); rodriguez_isacar@gva.es (I.R.-C.); martinez_belalc@gva.es (B.M.-A.); canet_rod@gva.es (R.C.)
* Correspondence: quinones_ana@gva.es

**Abstract:** The aim of this study was to establish DRIS (Diagnosis and Recommendation Integrated System) norms and Nutritional Optimal Ranges (NOR) for 'Rojo Brillante' Protected Designation of Origin (PDO) 'Ribera del Xúquer'. The database contained 800 leaf samples collected in different crop phenological stages [after flowering (AF), fruit enlargement (FE), fruit colouring (FC), and harvesting HV)]. DRIS norms (78) were established for macronutrients: N, P, K, Ca, Mg and S; micronutrients B, Cu, Fe, Mn, and Zn and salinity elements: Na and Cl. The Nutrient Balance Index (NBI; the absolute value of the sum of the DRIS indices) was used to determine the optimal sampling period. Fruit enlargement was the period during which persimmon trees were more nutritionally balanced regardless of sprout origin (vegetative or floral) and irrigation type (drip or flood) in orchards

**Keywords:** nutritional balance; diagnostic indices; mineral nutrition; persimmon; fertilisation





## 1. Introduction

Persimmon fruit (*Diospyros kaki* Thunb.) is one of the most important crops in the Mediterranean Region of Spain, which is the second biggest persimmon fruit producer in the world [1]. Among cultivars, 'Rojo Brillante' is the commonest variety, with the production of approximately 429,000 tons, in the Valencian Community and around 65,000 tons in Andalusia [2]. This fruit is grown in the Valencian Community along the Xúquer and Magro Rivers and has a Protected Designation of Origin (PDO) according to Regulation (EU) No. 1151/2012 [3], certified by the Regulatory Council of PDO 'Kaki Ribera del Xúquer' and accredited by the Spanish National Accreditation Body (ENAC). The 'Rojo Brillante' variety requires astringency elimination to be consumed. Traditionally, persimmon has been locally consumed because, in order to consume it, the fruit was harvested when it was overripe and had a soft gelatinous consistency. However, in 1997, postharvest technology began to be used to eliminate astringency, which is based on high $CO_2$ concentrations [4]. This enables them to be commercialised when they are very firm, and fruit can arrive at more distant markets.

As in other crops, one of the main objectives of persimmon production is to optimise yield with high fruit quality. The soil nutrient imbalance reduces the uptake of some nutrients, which makes the plants more sensitive to stress conditions [5,6]. The nutrient uptake of 'Rojo Brillante' persimmon fruit, PDO 'Kaki Ribera del Xúquer', has been estimated at 0.85 kg of nitrogen (N), 0.39 kg of $P_2O_5$ and 1.42 kg of $K_2O$ per ton of production [7]. Despite 'Rojo Brillante' being well-adapted to the climate conditions of the Mediterranean Region, the soil's ability to supply the necessary nutrients is insufficient to achieve adequate production and growth. Moreover, improving efficient irrigation and fertilisation management is considered crucial to enhance the conditions of this salinity-sensitive crop [8,9]. Therefore, fertilisation and irrigation must be sustainable practices that focus on minimising the environmental impact of agricultural production and land use and subsequent economic turnover. Traditionally persimmon crops have been irrigated by flooding, but

today drip irrigation is increasingly applied because it is recommended for improving water use efficiency and reducing the incidence of diseases [10].

Of all the tools available to optimise fertilisation, leaf nutritional analyses are essential. These analyses can be interpreted by different methods to determine optimal ranges of nutrients. Of them, the Diagnosis and Recommendation Integrated System (DRIS) provides a mathematical means to order a large number of nutrient ratios and/or products on easily interpretable nutrient indices [11]. Each relation between a pair of nutrients is determined as a DRIS norm. Once DRIS norms are determined, the DRIS index can be established for each nutrient using all the obtained DRIS norms that are related to the nutrient used to calculate the index.

The advantage of the DRIS method is that instead of determining the optimal value of each isolated nutrient, nutrient indices determine the balance of a specific nutrient based on the other considered nutrients. Due to that, the chances of making successful recommendations to apply the optimal doses of fertilisers increases. Nutrient indices indicate whether nutrients are balanced (zero or close to zero), in excess (positive indices) or deficient (negative indices). In persimmon crops, as in other crops, nutrient imbalances such as high potassium (K) and manganese (Mn) can limit calcium (Ca) uptake, which is needed especially in fruit growth stages I and II [12]. By setting DRIS indices to equal zero and taking into account the standard deviation (SD), the Nutrient Optimal Range (NOR) can be determined to establish reference tables of nutritional leaf profiles [13]. Thus, DRIS is a comprehensive system that identifies all the nutritional factors that limit crop production and, hence, increases the chances of obtaining high crop yields by improving fertiliser recommendations [14]. This method has been used with a wide range of crops (i.e., almond, mandarin, cotton or soya bean) from the time it was established to the present-day [15–18]. In persimmon crops, Klein et al. [19] in Israel and in a single phenological stage have employed the DRIS method. Their study determined the usefulness of this method for distinguishing the mineral profile of vigorous fruiting shoots from regular fruiting shoots based on the area and weight of leaves. Nevertheless, to improve the precision of this diagnosis, it is necessary to establish norms from regional and local studies that take into account the variability of the nutrients and crops studied (climatic characteristics, production level, etc.) [17]. The DRIS method has not been used for persimmon cv. 'Rojo Brillante' in the Mediterranean area previously.

Nutritional requirements vary according to plants' age and phenological stage. Thus, it is important to specify the optimal foliar sampling period for each species. Standards for the leaf analysis of fruit trees are usually developed during the minimal nutrient fluctuation period within the growing season, which usually occurs after shoot growth finishes [20]. In this sense, the DRIS method can be useful for determining it because the Nutrient Balance Index (NBI) is a tool derived from this method. The NBI is the absolute value of the sum of DRIS indices. The closer the NBI is to zero, the better plants' nutritional status is. Hence the NBI can help to know which phenological stage is more balanced.

In persimmon crops, different optimal periods have been recommended for leaf sampling. George et al. [21] advise carrying out two samplings; the first after fruit set when the least variation in leaf nutrient concentrations appears; the second 1 month before harvesting when critical leaf values are known and provide 90 % maximum fruit growth in the field experiments. Subsequently, Pomares et al. [7] indicate that sampling must be carried out during fruit enlargement (July in the Valencian Community) provided the period of stable leaf nutritional contents.

Both these recommendations have been made based on reference values when studying each nutrient individually. In studies based on the DRIS method in fruit trees, where the phenological plant stage is taken into account, different norms are determined for each nutrient ratio per specific period [22–25]. Nevertheless, an advantage of the DRIS method is that the database comprises the whole growing season.

The establishment of both norms and DRIS indices for persimmon cv. 'Rojo Brillante', in order to be able to improve fertiliser rates, is relevant in the current framework in

which regulations concerning agricultural practices are increasingly oriented towards the sustainability of agricultural resources. In the case of the European Commission, it will act to reduce nutrient losses by at least 50% while ensuring that soil fertility is not impaired. The objective of the European Commission is to reduce the use of fertilisers by at least 20% by 2030 [26].

Therefore, the main objectives of the present work are, in order to optimise the fertilisation in this crop, to establish the DRIS norms and DRIS indices to interpret the leaf nutritional status of cv. 'Rojo Brillante' persimmon to select the NOR of macro- and micronutrients in leaves and determine the adequate leaf sampling time regarding NBIs.

## 2. Materials and Methods

### 2.1. Site Description and Experimental Design

Experiments were carried out over two growing seasons (2017/2018 to 2018/2019) in 58 'Rojo Brillante' persimmon commercial orchards in the Valencia Region (East Spain) grafted onto *Diospyros lotus*. This region has a Mediterranean climate with 400–500 mm of mean annual rainfall and average annual temperatures of 15–17 °C. According to the World Reference Base for Soil Resources (WRB) classification system, the main soils used in this area for agriculture are Fluvisols and Antrosols [27].

Trees received an annual supply of fertilisers, which provided 250–220/100–75/135/25 UF (kg·ha$^{-1}$) N/P$_2$O$_5$/K$_2$O/MgO for flood and drip irrigation, respectively, according to the cultural practices applied in this agricultural area. To obtain representative samples of the growing conditions in this area, the orchards that were irrigated by flood and drip irrigation systems were selected.

### 2.2. Leaf Sampling

Leaves were collected from both vegetative and reproductive shoots to know if behaviour differed between both flushes. Upon sampling, 60 leaves were randomly sampled around the canopy (N-S-E-W) in each orchard. Although there is a general consensus that the leaf is the ideal tissue to be sampled to diagnose plant nutrient status, this opinion is divided as to whether the petiole, lamina or whole leaf should be used. In these studies, the whole leaf was employed because persimmon is a temperate tree crop [28]. In order to study the effect of phenological stage, sampling was carried out on the following BBCH scale [29]:

- After flowering (AF): Growth stage 65–67. From full flowering (50% of flowers are open) to flower fading (most petals fall or dry up);
- Fruit enlargement (FE): Growth stage 74–77. Fruit is about 40–70% of its final size;
- Fruit colouring (FC): Growth stage 81. Initiation of skin colour change;
- Harvest period (HV): Growth stage 85–87. From advanced ripening to developed fruit colour fruit and fruit is ripe for consumption (only not astringency cultivars).

### 2.3. Leaf Sample Preparation and Analysis Methodology

All the leaf samples were carefully washed with deionised water, dried in a forced-air oven at 70 °C for 48 h and ground to 1 mm in a water-refrigerated mill (IKA M 20, IKA Labortechnik, Staufen, Germany). Then the analysis was carried out to determine salinity [chloride (Cl) and sodium (Na), macronutrients (N, phosphorus (P), K, Ca, magnesium (Mg) and sulphur S)] and micronutrients [boron (B), copper (Cu), iron (Fe), Mn and zinc (Zn)]. N was determined according to the Kjeldahl method using an Elemental Analyser (NC2500 Thermo Finnigan, Bremen, Germany). Total Cl ion content was established by argentometry in a chloridometer (Corning Ltd., Halstead Essex, UK) in acetic acid 2% extracts obtained by shaking the dried leaf material for 24 h [30]. The determination of the remaining elements was made in the leaf extracts obtained by an open-vessel procedure [31]. This involved the overnight predigestion of 0.5 g of dried plant material with 10 mL of concentrated HNO$_3$, followed by digestion at 120 °C, then cooling before adding 2 mL of ultratrace metal grade (70% HClO$_4$), and final digestion at 220 °C until white fumes appear. The thus obtained digest was diluted with 25 mL of ultrapure water before determining the elements

by simultaneous inductively coupled plasma atomic emission spectrometry (ICAP-AES 6000, Thermo Scientific, Cambridge, UK). The macronutrient concentration results were expressed as g nutrient·100 g dry weight$^{-1}$ (%) and the micronutrient concentration as mg nutrient·1000 g dry weight$^{-1}$ (ppm).

### 2.4. DRIS Norms

For establishing DRIS norms, data were divided into two subpopulations based on commercial yields: high-yielding (reference) and low-yielding (non-reference) populations. The high-yielding population comprised the orchards with a persimmon yield over the mean + 0.5·SD. DRIS norms were obtained according to the procedure by Walworth and Sumner [11]. In order to determine DRIS norms for each pair of nutrients, three forms of expression can be considered: X/Y, Y/X, or X·Y. The dual nutrient ratio order that exhibited the widest variance ($\sigma^2$) of the low-yielding group to that of the high-yielding population was selected as norm because a low-yielding population is expected to have wider variance for being more imbalanced. The mean value of each ratio in the high-yielding population was used as DRIS norms.

### 2.5. DRIS Indices

DRIS indices were calculated once norms were determined. According to Jones [32], to simplify the logic and calculation of DRIS in agreement with Beaufils [33], the intermediate DRIS function values of each sample from the average values of the same ratios in the reference population are determined as follows:

$$f\left(\frac{Y}{X}\right) = \left[\left(\frac{Y}{X}\right) - \left(\frac{y}{x}\right)\right] \times \left(\frac{c}{SD}\right) \tag{1}$$

where:

$f\left(\frac{Y}{X}\right)$ is the calculated function for nutrient ratios X and Y;

$\left(\frac{Y}{X}\right)$ is the nutrient relation of the sample;

$\left(\frac{y}{x}\right)$ is the nutrient relation of the standard norm obtained with the reference population;

$c$ is a constant equivalent to 1, and SD is the standard deviation of the standard norm.

The obtained functions are then used to generate DRIS indices by the following equation applied to each nutrient element:

$$Iy = \frac{\left[\sum_{i=1}^{m} f\left(\frac{Y}{Xi}\right) - \sum_{i=1}^{n} f\left(\frac{Zi}{Y}\right)\right]}{m + n} \tag{2}$$

where:

$Iy$ is the DRIS index for the y nutrient;

$\sum_{i=1}^{m} f\left(\frac{Y}{Xi}\right)$ is the value of the total functions where the nutrient element of the Y index is in the numerator of the DRIS norms;

And $\sum_{i=1}^{n} f\left(\frac{Zi}{Y}\right)$ is the value of the total functions where the nutrient element of the Y index is in the denominator and $m + n$ are the number of used functions.

### 2.6. The Nutrient Optimal Range (NOR)

The NOR was determined by taking the mean nutrient concentration value of the high-yielding population. Five classes of nutrient ranges were determined in the different sampled phenological stages by taking into account SD [34]:

- Nutritional deficit status < −4/3 SD
- Deficiency tendency = −4/3 SD to −2/3 SD;
- Sufficiency = −2/3 SD to 2/3 SD;
- Excess tendency = 2/3 SD to 4/3 SD;
- Excessive nutritional status > 4/3 SD.

### 2.7. Determining the Sampling Period for the Nutritional Diagnosis

The NBI was calculated after the nutrient DRIS indices. In this study, the NBI was determined in each phenological stage by taking into account yield, irrigation system and leaf flush (vegetative and reproductive flushes). The NBI was calculated by adding the absolute value of the DRIS indices of each sample according to the following equation:

$$NBI = \sum_{i=1}^{m} |I_i + \ldots + I_m| \tag{3}$$

### 2.8. Statistical Analyses

All the statistical analyses were performed using Statgraphics Centurion (version 5.1; Statpoint Technologies, Warrenton, VA, USA). The relation between leaf nutrient content and DRIS indices was studied by regression analysis, and it was shown in a simple scatter plot. The nutrient leaf content was the independent variable, and DRIS index was the dependent variable. Among all possible linear and non-linear regressions, the one with the highest correlation coefficient ($R^2$) value has been selected.

## 3. Results and Discussion

The study was conducted to establish the nutritional balance of 12 macro- and micronutrients in 'Rojo Brillante' persimmon, PDO 'Kaki Ribera del Xúquer' Spain). The high-yielding population was made up of 13 of the 58 orchards, with yields from 46.38 t·ha$^{-1}$ to 81.32 t·ha$^{-1}$.

### 3.1. DRIS Norms

The orchards with yields higher than 46.38 t·ha$^{-1}$ were considered high-yielding, and they represented 22.41% of the entire dataset.

The mean nutrient ratios and SDs of the high-yielding population are shown in Table 1.

**Table 1.** Diagnosis and Recommendation Integrated System (DRIS) norms and standard deviations (SD) obtained from the nutrient contents of foliar samples from a high-yielding population of 'Rojo Brillante' persimmon, PDO 'Kaki Ribera del Xúquer', Spain.

| | Nutrient | Mean | SD | | Norm | Mean | SD | | Norm | Mean | SD |
|---|---|---|---|---|---|---|---|---|---|---|---|
| | N (%) | 1.837 | 0.255 | 17 | P/B | 0.003 | 0.003 | 48 | Mg/Zn | 0.017 | 0.009 |
| | P (%) | 0.106 | 0.019 | 18 | P/Cu | 0.028 | 0.012 | 49 | Fe/Mg | 106.796 | 40.603 |
| | K (%) | 1.401 | 0.311 | 19 | P/Fe | 0.002 | 0.001 | 50 | Mn/Mg | 321.806 | 245.416 |
| | Ca (%) | 1.832 | 0.827 | 20 | K/P | 13.252 | 2.485 | 51 | Na/S | 0.063 | 0.025 |
| | Mg (%) | 0.530 | 0.184 | 21 | Na/P | 0.133 | 0.052 | 52 | Na/Cl | 0.022 | 0.015 |
| | Na (%) | 0.014 | 0.005 | 22 | Mn/P | 1689.810 | 1398.100 | 53 | Na/B | 0.0004 | 0.0002 |
| | S (%) | 0.225 | 0.046 | 23 | Zn/P | 353.202 | 147.619 | 54 | Na/Cu | 0.004 | 0.002 |
| | Cl (%) | 0.808 | 0.400 | 24 | K/Ca | 1.009 | 0.665 | 55 | Na/Mn | 0.0001 | 0.0001 |
| | B (ppm) | 46.728 | 26.832 | 25 | K/Mg | 3.117 | 1.571 | 56 | Na/Zn | 0.0004 | 0.0002 |
| | Cu (ppm) | 4.704 | 2.697 | 26 | K/Na | 115.132 | 51.830 | 57 | Fe/Na | 4104.840 | 1700.294 |
| | Fe (ppm) | 55.539 | 19.712 | 27 | K/S | 6.478 | 1.720 | 58 | S/Cl | 0.364 | 0.233 |
| | Mn (ppm) | 52.806 | 15.844 | 28 | K/Cl | 2.467 | 1.977 | 59 | S/B | 0.007 | 0.006 |
| | Zn (ppm) | 35.603 | 13.301 | 29 | K/B | 0.044 | 0.035 | 60 | S/Cu | 0.059 | 0.028 |
| | | | | 30 | K/Cu | 0.370 | 0.165 | 61 | S/Fe | 0.005 | 0.002 |
| | Norm | Mean | SD | 31 | K/Zn | 0.045 | 0.023 | 62 | S/Zn | 0.007 | 0.003 |
| 1 | N/Na | 149.086 | 69.8038 | 32 | Fe/K | 39.738 | 15.495 | 63 | Mn/S | 821.265 | 717.909 |
| 2 | N/S | 8.468 | 1.891 | 33 | Mn/K | 135.820 | 122.740 | 64 | Cl/B | 0.020 | 0.012 |
| 3 | N/Cl | 3.169 | 2.322 | 34 | Ca/Mg | 3.344 | 0.834 | 65 | Cl/Cu | 0.232 | 0.177 |
| 4 | N/B | 0.060 | 0.046 | 35 | Ca/Cl | 2.387 | 0.762 | 66 | Cl/Mn | 0.008 | 0.007 |
| 5 | N/Cu | 0.495 | 0.210 | 36 | Ca/B | 0.046 | 0.026 | 67 | Fe/Cl | 82.127 | 49.408 |
| 6 | N/Fe | 0.039 | 0.015 | 37 | Ca/Cu | 0.510 | 0.339 | 68 | Zn/Cl | 54.806 | 37.647 |
| 7 | N/Zn | 0.057 | 0.024 | 38 | Na/Ca | 0.009 | 0.005 | 69 | B/Cu | 12.442 | 8.851 |
| 8 | Ca/N | 1.060 | 0.573 | 39 | S/Ca | 0.157 | 0.098 | 70 | B/Zn | 1.461 | 1.061 |
| 9 | K/N | 0.765 | 0.169 | 40 | Fe/Ca | 34.013 | 17.862 | 71 | Fe/B | 1.453 | 0.915 |
| 10 | Mg/N | 0.303 | 0.132 | 41 | Mn/Ca | 95.052 | 71.053 | 72 | Mn/B | 3.824 | 2.735 |

**Table 1.** *Cont.*

|  | Nutrient | Mean | SD |  | Norm | Mean | SD |  | Norm | Mean | SD |
|----|----------|-------|--------|----|-------|--------|--------|----|-------|--------|--------|
| 11 | P/N | 0.058 | 0.009 | 42 | Zn/Ca | 23.796 | 15.013 | 73 | Fe/Cu | 15.117 | 7.301 |
| 12 | Mn/N | 80.440 | 63.845 | 43 | Mg/Na | 41.094 | 16.650 | 74 | Mn/Cu | 46.064 | 42.338 |
| 13 | P/Ca | 0.078 | 0.061 | 44 | Mg/S | 2.424 | 0.836 | 75 | Zn/Cu | 9.448 | 5.431 |
| 14 | P/Mg | 0.231 | 0.124 | 45 | Mg/Cl | 0.768 | 0.337 | 76 | Fe/Mn | 0.487 | 0.354 |
| 15 | P/S | 0.489 | 0.096 | 46 | Mg/B | 0.015 | 0.009 | 77 | Fe/Zn | 1.727 | 0.788 |
| 16 | P/Cl | 0.184 | 0.161 | 47 | Mg/Cu | 0.145 | 0.085 | 78 | Mn/Zn | 4.455 | 3.675 |

The variance ratios of all the selected dual ratios of the low- vs. high-yielding population were higher than 1, which indicates a relatively higher variance for the low-yielding population (data not shown).

Table 1 shows a wide range between the obtained ratios, from values per $10^3$ to values of $10^{-3}$. This is because a decision was made to maintain macronutrients as percentages and micronutrients as parts per million, which are frequently used units. This was performed to confer more practicality to the use of the norms obtained by farmers in the future. After obtaining DRIS norms, the equations to determine the DRIS indices of the selected nutrients were as follows (Table 2):

Figures 1 and 2 show the relation between the leaf nutrient concentrations of the high-yielding population and DRIS indices, estimated by regression analysis. In all cases, the lower the DRIS index, the lower the nutrient concentration. This increases the reliability of DRIS norms because approaches correctly indicate nutritional limitations [16]. For each relation, the regression models that were best adapted to data evolution were fitted depending on the percentage of variation of the dependent variable that was collectively explained by all the independent variables ($R^2$). In DRIS studies, linear regression models are often fitted to the relation between the DRIS index and leaf nutrient concentration [13,35]. Other authors have established that linear, quadratic and logarithmic are the best fitting regression models [16,36,37]. Nevertheless, in this study, a wide range of regression models was selected as each relationship between the nutrient concentration and its DRIS index differently evolved depending on how the plant vegetative cycle advanced. Thus a single linear regression model was used for N, while non-linear regression models were selected for the other nutrients. Of the non-linear models, five exponential regression models (Ca, S, Na, B and Zn), four square root-X (K, Mg, Fe and Mn), two inverse Y (Cl and Cu), and one Y quadratic (P) were obtained.

**Table 2.** DRIS index equations of macro- and micronutrients according to Jones [32], where letters of macro- and micronutrients are related to the sample to analyse elements.

**DRIS Indices**

$$I_N = \left[\left[\left(\frac{\frac{N}{Na}-149.086}{69.838}\right) + \left(\frac{\frac{N}{S}-8.468}{1.891}\right) + \left(\frac{\frac{N}{Cl}-3.169}{2.322}\right) + \left(\frac{\frac{N}{B}-0.060}{0.046}\right) + \left(\frac{\frac{N}{Cu}-0.495}{0.210}\right) + \left(\frac{\frac{N}{Fe}-0.039}{0.015}\right) + \left(\frac{\frac{N}{Zn}-0.057}{0.024}\right)\right] - \left[\left(\frac{\frac{Ca}{N}-1.060}{0.573}\right) + \left(\frac{\frac{K}{N}-0.765}{0.169}\right) + \left(\frac{\frac{Mg}{N}-0.303}{0.132}\right) + \left(\frac{\frac{P}{N}-0.058}{0.009}\right) + \left(\frac{\frac{Mn}{N}-80.44}{63.845}\right)\right]\right] \times \frac{1}{12}$$

$$I_P = \left[\left[\left(\frac{\frac{P}{N}-0.058}{0.009}\right) + \left(\frac{\frac{P}{Ca}-0.078}{0.061}\right) + \left(\frac{\frac{P}{Mg}-0.231}{0.124}\right) + \left(\frac{\frac{P}{S}-0.489}{0.096}\right) + \left(\frac{\frac{P}{Cl}-0.184}{0.161}\right) + \left(\frac{\frac{P}{B}-0.003}{0.003}\right) + \left(\frac{\frac{P}{Cu}-0.028}{0.012}\right) + \left(\frac{\frac{P}{Fe}-0.002}{0.001}\right)\right] - \left[\left(\frac{\frac{K}{P}-13.252}{2.485}\right) + \left(\frac{\frac{Na}{P}-0.133}{0.052}\right) + \left(\frac{\frac{Mn}{P}-1689.81}{1398.1}\right) + \left(\frac{\frac{Zn}{P}-352.202}{147.619}\right)\right]\right] \times \frac{1}{12}$$

$$I_K = \left[\left[\left(\frac{\frac{K}{N}-0.765}{0.169}\right) + \left(\frac{\frac{K}{P}-13.252}{2.485}\right) + \left(\frac{\frac{K}{Ca}-3.169}{2.322}\right) + \left(\frac{\frac{K}{Mg}-3.117}{1.571}\right) + \left(\frac{\frac{K}{Na}-115.132}{51.830}\right) + \left(\frac{\frac{K}{S}-6.478}{1.720}\right) + \left(\frac{\frac{K}{Cl}-2.467}{1.977}\right) + \left(\frac{\frac{K}{B}-0.044}{0.035}\right) + \left(\frac{\frac{K}{Cu}-0.370}{0.165}\right) + \left(\frac{\frac{K}{Zn}-0.045}{0.023}\right)\right] - \left[\left(\frac{\frac{Fe}{K}-39.738}{15.495}\right) + \left(\frac{\frac{Mn}{K}-135.82}{122.74}\right)\right]\right] \times \frac{1}{12}$$

$$I_{Ca} = \left[\left[\left(\frac{\frac{Ca}{N}-1.060}{0.573}\right) + \left(\frac{\frac{Ca}{Mg}-3.344}{0.834}\right) + \left(\frac{\frac{Ca}{Cl}-2.387}{0.762}\right) + \left(\frac{\frac{Ca}{B}-0.046}{0.026}\right) + \left(\frac{\frac{Ca}{Cu}-0.510}{0.339}\right)\right] - \left[\left(\frac{\frac{P}{Ca}-0.078}{0.061}\right) + \left(\frac{\frac{K}{Ca}-1.009}{0.665}\right) + \left(\frac{\frac{Na}{Ca}-0.009}{0.005}\right) + \left(\frac{\frac{S}{Ca}-0.157}{0.098}\right) + \left(\frac{\frac{Fe}{Ca}-34.013}{17.862}\right) + \left(\frac{\frac{Mn}{Ca}-95.052}{71.053}\right) + \left(\frac{\frac{Zn}{Ca}-23.796}{15.013}\right)\right]\right] \times \frac{1}{12}$$

$$I_{Mg} = \left[\left[\left(\frac{\frac{Mg}{N}-0.303}{0.132}\right) + \left(\frac{\frac{Mg}{Na}-41.094}{16.650}\right) + \left(\frac{\frac{Mg}{S}-2.424}{0.836}\right) + \left(\frac{\frac{Mg}{Cl}-0.768}{0.337}\right) + \left(\frac{\frac{Mg}{B}-0.015}{0.009}\right) + \left(\frac{\frac{Mg}{Cu}-0.145}{0.085}\right) + \left(\frac{\frac{Mg}{Zn}-0.017}{0.009}\right)\right] - \left[\left(\frac{\frac{P}{Mg}-0.231}{0.124}\right) + \left(\frac{\frac{K}{Mg}-3.117}{1.571}\right) + \left(\frac{\frac{Ca}{Mg}-3.344}{0.834}\right) + \left(\frac{\frac{Fe}{Mg}-106.796}{40.603}\right) + \left(\frac{\frac{Mn}{Mg}-321.806}{245.416}\right)\right]\right] \times \frac{1}{12}$$

$$I_{Na} = \left[\left[\left(\frac{\frac{Na}{P}-0.133}{0.052}\right) + \left(\frac{\frac{Na}{Ca}-0.009}{0.005}\right) + \left(\frac{\frac{Na}{S}-0.063}{0.025}\right) + \left(\frac{\frac{Na}{Cl}-0.022}{0.015}\right) + \left(\frac{\frac{Na}{B}-0.0004}{0.0002}\right) + \left(\frac{\frac{Na}{Cu}-0.004}{0.002}\right) + \left(\frac{\frac{Na}{Mn}-0.0001}{0.0001}\right) + \left(\frac{\frac{Na}{Zn}-0.0004}{0.0002}\right)\right] - \left[\left(\frac{\frac{N}{Na}-149.086}{69.838}\right) + \left(\frac{\frac{K}{Na}-115.132}{51.830}\right) + \left(\frac{\frac{Mg}{Na}-41.094}{16.650}\right) + \left(\frac{\frac{Fe}{Na}-4104.84}{1700.294}\right)\right]\right] \times \frac{1}{12}$$

$$I_S = \left[\left[\left(\frac{\frac{S}{Ca}-0.157}{0.098}\right) + \left(\frac{\frac{S}{Cl}-0.364}{0.233}\right) + \left(\frac{\frac{S}{B}-0.007}{0.006}\right) + \left(\frac{\frac{S}{Cu}-0.059}{0.028}\right) + \left(\frac{\frac{S}{Fe}-0.005}{0.002}\right) + \left(\frac{\frac{S}{Zn}-0.007}{0.003}\right)\right] - \left[\left(\frac{\frac{N}{S}-8.468}{1.891}\right) + \left(\frac{\frac{P}{S}-0.489}{0.096}\right) + \left(\frac{\frac{K}{S}-6.478}{1.720}\right) + \left(\frac{\frac{Mg}{S}-2.424}{0.836}\right) + \left(\frac{\frac{Na}{S}-0.063}{0.025}\right) + \left(\frac{\frac{Mn}{S}-821.265}{717.909}\right)\right]\right] \times \frac{1}{12}$$

$$I_{Cl} = \left[\left[\left(\frac{\frac{Cl}{B}-0.020}{0.012}\right) + \left(\frac{\frac{Cl}{Cu}-0.232}{0.177}\right) + \left(\frac{\frac{Cl}{Mn}-0.008}{0.007}\right)\right] - \left[\left(\frac{\frac{N}{Cl}-3.169}{2.322}\right) + \left(\frac{\frac{P}{Cl}-0.184}{0.161}\right) + \left(\frac{\frac{K}{Cl}-2.467}{1.977}\right) + \left(\frac{\frac{Ca}{Cl}-2.387}{0.762}\right) + \left(\frac{\frac{Mg}{Cl}-0.768}{0.337}\right) + \left(\frac{\frac{Na}{Cl}-0.022}{0.015}\right) + \left(\frac{\frac{S}{Cl}-0.364}{0.233}\right) + \left(\frac{\frac{Fe}{Cl}-82.127}{49.408}\right) + \left(\frac{\frac{Zn}{Cl}-54.806}{37.647}\right)\right]\right] \times \frac{1}{12}$$

$$I_B = \left[\left[\left(\frac{\frac{B}{Cu}-12.422}{8.851}\right) + \left(\frac{\frac{B}{Zn}-1.461}{1.061}\right)\right] - \left[\left(\frac{\frac{N}{B}-0.060}{0.046}\right) + \left(\frac{\frac{P}{B}-0.003}{0.003}\right) + \left(\frac{\frac{K}{B}-0.044}{0.035}\right) + \left(\frac{\frac{Ca}{B}-0.046}{0.026}\right) + \left(\frac{\frac{Mg}{B}-0.015}{0.009}\right) + \left(\frac{\frac{Na}{B}-0.0004}{0.0002}\right) + \left(\frac{\frac{S}{B}-0.007}{0.006}\right) + \left(\frac{\frac{Cl}{B}-0.020}{0.012}\right) + \left(\frac{\frac{Fe}{B}-1.453}{0.915}\right) + \left(\frac{\frac{Mn}{B}-3.824}{2.735}\right)\right]\right] \times \frac{1}{12}$$

$$I_{Cu} = -\left[\left(\frac{\frac{N}{Cu}-0.495}{0.210}\right) + \left(\frac{\frac{P}{Cu}-0.028}{0.0121}\right) + \left(\frac{\frac{K}{Cu}-0.370}{0.165}\right) + \left(\frac{\frac{Ca}{Cu}-0.510}{0.339}\right) + \left(\frac{\frac{Mg}{Cu}-0.145}{0.085}\right) + \left(\frac{\frac{Na}{Cu}-0.004}{0.002}\right) + \left(\frac{\frac{S}{Cu}-0.059}{0.028}\right) + \left(\frac{\frac{Cl}{Cu}-0.232}{0.177}\right) + \left(\frac{\frac{B}{Cu}-12.442}{8.851}\right) + \left(\frac{\frac{Fe}{Cu}-15.117}{7.301}\right) + \left(\frac{\frac{Mn}{Cu}-46.064}{42.338}\right) + \left(\frac{\frac{Zn}{Cu}-9.448}{5.431}\right)\right] \times \frac{1}{12}$$

$$I_{Fe} = \left[\left[\left(\frac{\frac{Fe}{K}-39.798}{15.495}\right) + \left(\frac{\frac{Fe}{Ca}-34.013}{17.862}\right) + \left(\frac{\frac{Fe}{Mg}-106.796}{40.603}\right) + \left(\frac{\frac{Fe}{Na}-4104.84}{1700.294}\right) + \left(\frac{\frac{Fe}{Cl}-82.127}{49.408}\right) + \left(\frac{\frac{Fe}{B}-1.453}{0.915}\right) + \left(\frac{\frac{Fe}{Cu}-15.117}{7.301}\right) + \left(\frac{\frac{Fe}{Mn}-0.487}{0.354}\right) + \left(\frac{\frac{Fe}{Zn}-1.727}{0.788}\right)\right] - \left[\left(\frac{\frac{N}{Fe}-0.039}{0.015}\right) + \left(\frac{\frac{P}{Fe}-0.002}{0.001}\right) + \left(\frac{\frac{S}{Fe}-0.005}{0.002}\right)\right]\right] \times \frac{1}{12}$$

$$I_{Mn} = \left[\left[\left(\frac{\frac{Mn}{N}-80.44}{63.845}\right) + \left(\frac{\frac{Mn}{P}-1689.81}{1398.1}\right) + \left(\frac{\frac{Mn}{K}-135.82}{122.74}\right) + \left(\frac{\frac{Mn}{Ca}-95.052}{71.053}\right) + \left(\frac{\frac{Mn}{Mg}-321.806}{245.416}\right) + \left(\frac{\frac{Mn}{S}-821.265}{717.909}\right) + \left(\frac{\frac{Mn}{B}-3.824}{2.735}\right) + \left(\frac{\frac{Mn}{Cu}-46.064}{42.338}\right) + \left(\frac{\frac{Mn}{Zn}-4.455}{3.675}\right)\right] - \left[\left(\frac{\frac{Na}{Mn}-0.0001}{0.0001}\right) + \left(\frac{\frac{Cl}{Mn}-0.008}{0.007}\right) + \left(\frac{\frac{Fe}{Mn}-0.487}{0.354}\right)\right]\right] \times \frac{1}{12}$$

$$I_{Zn} = \left[\left[\left(\frac{\frac{Zn}{P}-353.202}{147.619}\right) + \left(\frac{\frac{Zn}{Ca}-23.796}{15.013}\right) + \left(\frac{\frac{Zn}{Cl}-54.806}{37.647}\right) + \left(\frac{\frac{Zn}{Cu}-9.448}{5.431}\right)\right] - \left[\left(\frac{\frac{N}{Zn}-0.057}{0.024}\right) + \left(\frac{\frac{K}{Zn}-0.045}{0.023}\right) + \left(\frac{\frac{Mg}{Zn}-0.017}{0.009}\right) + \left(\frac{\frac{Na}{Zn}-0.0004}{0.0002}\right) + \left(\frac{\frac{S}{Zn}-0.007}{0.003}\right) + \left(\frac{\frac{B}{Zn}-1.461}{1.061}\right) + \left(\frac{\frac{Fe}{Zn}-1.727}{0.788}\right) + \left(\frac{\frac{Mn}{Zn}-4.455}{3.675}\right)\right]\right] \times \frac{1}{12}$$

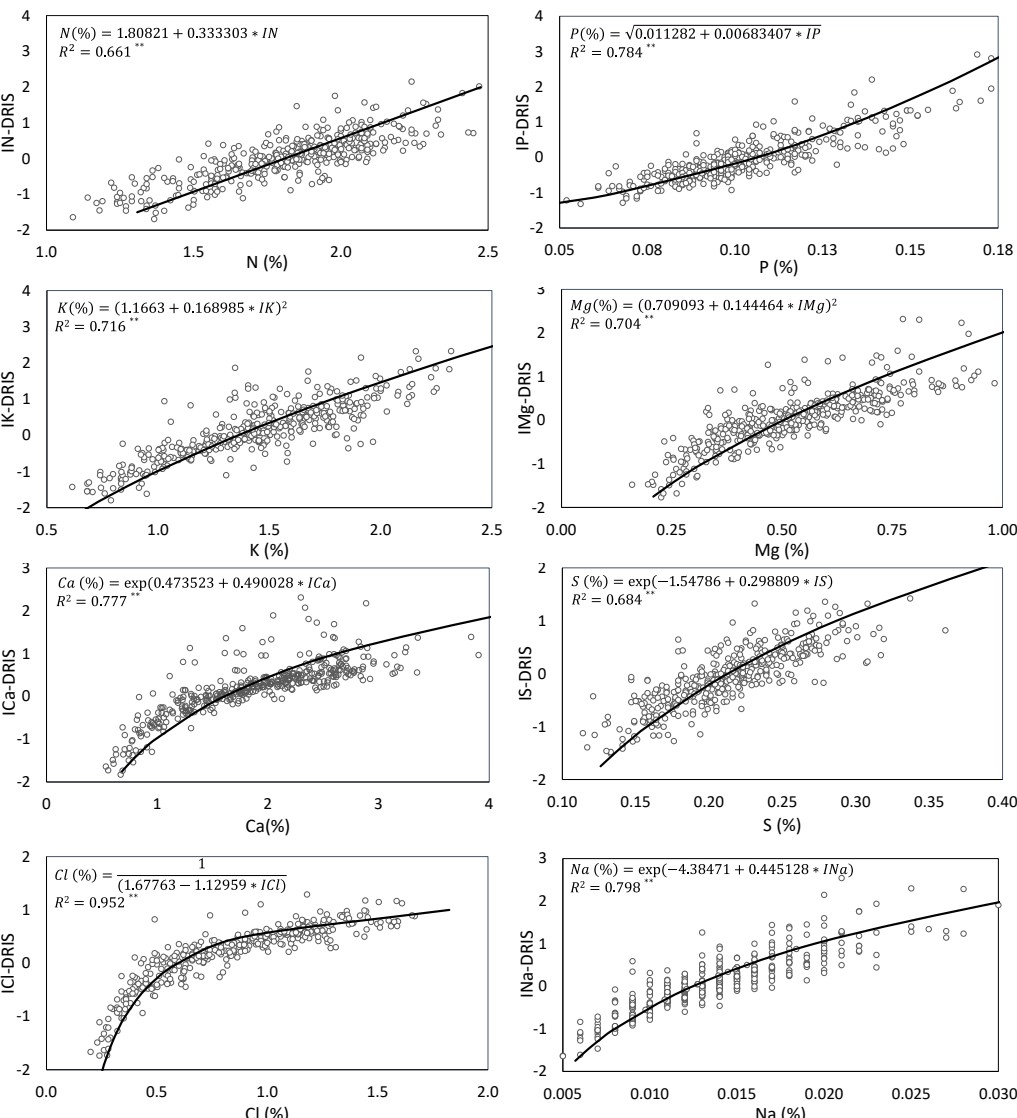

**Figure 1.** Relation between the leaf macronutrient concentrations in the high-yielding population and their corresponding nutrient DRIS indices (** indicates *p* < 0.01).

The DRIS index values well explained the leaf nutrient concentrations because all the obtained regression equations had an $R^2$ > 0.6 (*p* < 0.01). These results confirm that the DRIS method effectively diagnoses the nutritional status of persimmon crops. The DRIS indices of micronutrients, especially Cu and Mn ($R^2$ > 0.94), more clearly explained the leaf nutrient concentrations than the DRIS indices of macronutrients. The higher $R^2$ in micronutrients than in macronutrients has been previously reported in 'Gigante' cactus pear and cotton [16,38]. Of the macronutrients, N and S showed the weakest correlation with their DRIS indices ($R^2$ < 0.7). In other crops, a low correlation of the N and S concentrations with their respective DRIS indices has been observed in grass, apple and corn. This implies that other important factors play a role in the nutrient balance [36,39,40]. Aliyu et al. [35] have also observed this in corn, who attributed the low $R^2$ in the relation of the N concentration to the IN with N imbalance, instead of N deficiency, provided the S and Zn additions in their study.

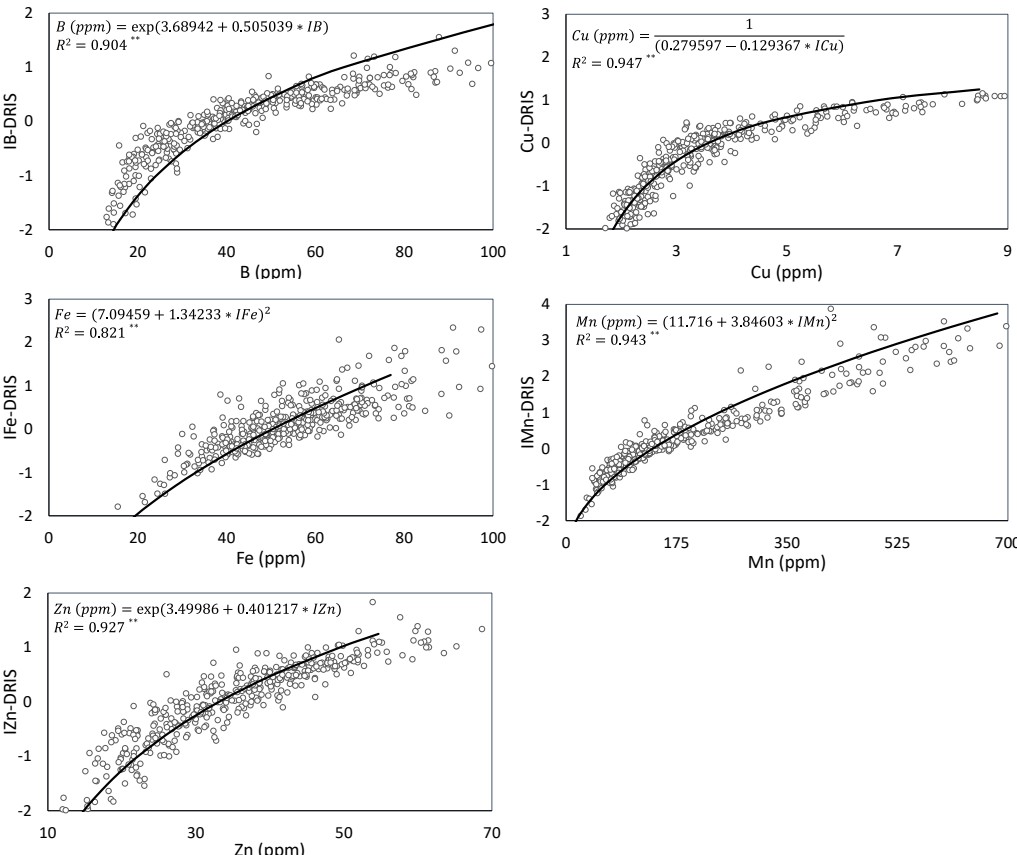

**Figure 2.** Relation between the leaf micronutrient concentrations in the high-yielding population and their corresponding nutrient DRIS indices (** indicates $p < 0.01$).

Based on the relation between the DRIS indices and nutrient contents in the leaf samples of the high-yielding population, it was possible to determine NORs. A plant is nutritionally balanced when DRIS indices come close to zero [11]. NORs can be obtained when the regression models of the relation of nutrient content and indices move towards this nutritional balance point (BP) [13].

For this study, Figures 3 and 4 represent the optimal range of each nutrient in relation to the estimated DRIS index in the different evaluated phenological stages (AF, FE, FC, HV) and the BP when regression equations equal zero.

As the value of each ratio is added to one index subtotal and subtracted from another before averaging, all the DRIS indices are balanced at around zero [11]. Accordingly, having obtained the $I_{DRIS}$ equations, the equilibrium point of each nutrient was determined by setting each $I_{DRIS}$ equation to equal zero. In most cases, the BP did not fall within the optimal range of all the phenological stages. Only for S and Zn was the BP within the optimal range of the concentration of these nutrients in all four studied phenological stages. Despite not finding the BP in all the phenological stages in the other nutrients, the triangle was found in the phenological FE stage in them all. Only for N was the BP not located in FE, but it was on the vertical line, which indicates the optimal lower range of this stage.

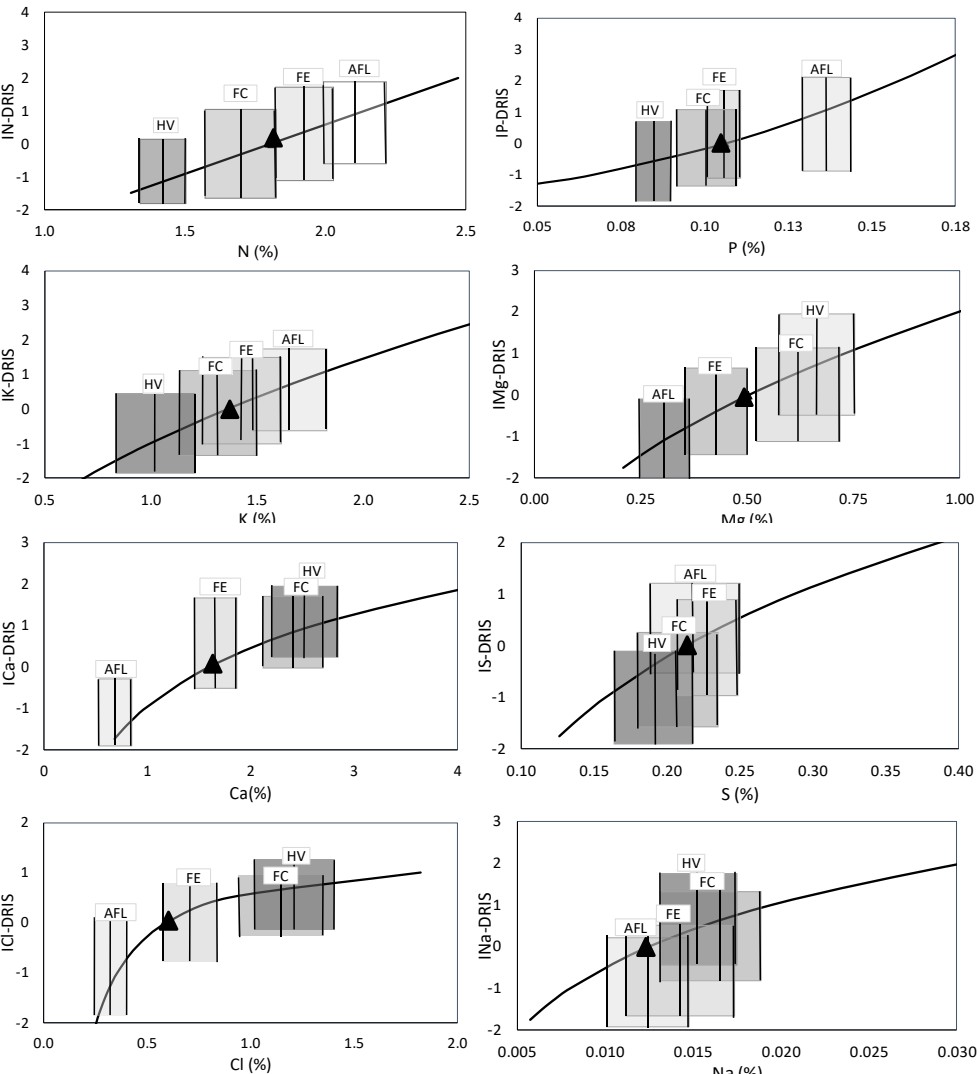

**Figure 3.** Optimal range of leaf macronutrients in four phenological stages: AF (After flowering), FE (fruit enlargement), FC (fruit colouring) and HV (harvest period). The vertical bar in the middle of each square represents the mean value of each phenological stage. The left vertical bar denotes the lower limit of the optimal range (−2/3 SD). The right bar indicates the upper limit of the optimal range (+2/3 SD). A black triangle represents the BP ($I_{DRIS} = 0$).

These results indicated that FE could be the best-recommended stage for carrying out leaf analyses to correct fertilisation programmes (nutrient doses). However, provided the importance of knowing the optimal foliar sampling time, an in-depth study was carried out, as mentioned in the subsequent section.

Moreover, to evaluate any statistical differences among the different sampling times, the mean nutrient value for 'Rojo Brillante' persimmon was taken from the samples of the high-yielding population in each determined phenological stage (AFL, FE, FC, HV) (Table 3).

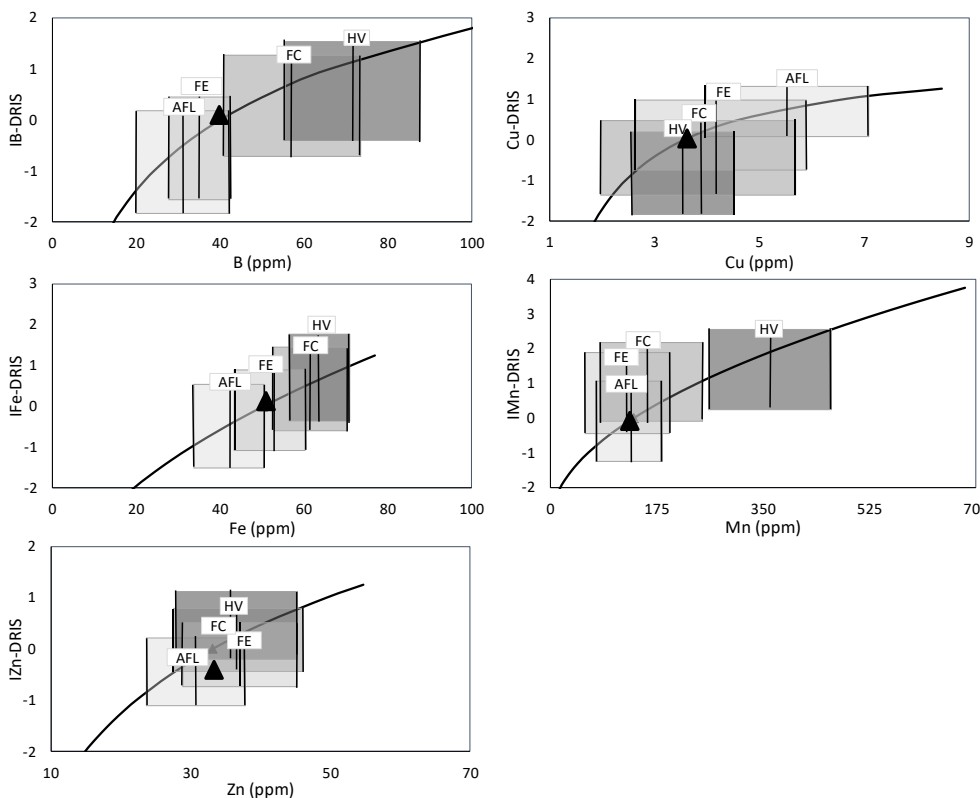

**Figure 4.** Optimal range of leaf micronutrients in four phenological stages: AF, FE, FC and HV. The vertical bar in the middle of each square represents the mean value of each phenological stage. The left vertical bar denotes the lower limit of the optimal range ($-2/3$ SD). The right bar indicates the upper limit of the optimal range ($+2/3$ SD). A black triangle represents the BP ($I_{DRIS} = 0$).

**Table 3.** Mean values and standard deviation (SD) of the foliar nutrient concentration in the different phenological stages for persimmon fruit PDO 'Kaki Ribera del Xúquer', Spain.

|  | After Flowering | | Fruit Enlargement | | Fruit Colouring | | Harvesting | |
|---|---|---|---|---|---|---|---|---|
|  | Mean | SD | Mean | SD | Mean | SD | Mean | SD |
| N (%) | 2.10 d | 0.17 | 1.93 c | 0.15 | 1.70 b | 0.19 | 1.42 a | 0.12 |
| P (%) | 0.135 c | 0.011 | 0.104 b | 0.006 | 0.100 b | 0.014 | 0.084 a | 0.007 |
| K (%) | 1.62 d | 0.27 | 1.44 c | 0.18 | 1.38 b | 0.34 | 1.07 a | 0.25 |
| Ca (%) | 0.75 a | 0.27 | 1.75 b | 0.24 | 2.62 c | 0.60 | 2.45 c | 0.25 |
| Mg (%) | 0.31 a | 0.08 | 0.54 b | 0.09 | 0.66 c | 0.16 | 0.65 c | 0.11 |
| Na (%) | 0.011 a | 0.003 | 0.014 b | 0.004 | 0.018 c | 0.005 | 0.015 bc | 0.003 |
| S (%) | 0.22 d | 0.03 | 0.24 c | 0.04 | 0.21 bc | 0.04 | 0.19 a | 0.03 |
| Cl (%) | 0.30 a | 0.14 | 0.72 b | 0.13 | 1.24 c | 0.18 | 1.20 c | 0.15 |
| B (ppm) | 39.80 a | 20.08 | 32.81 a | 12.62 | 58.00 b | 18.44 | 62.27 c | 32.74 |
| Cu (ppm) | 5.53 c | 2.35 | 3.79 b | 1.60 | 3.04 ab | 1.10 | 3.53 a | 1.49 |
| Fe (ppm) | 42.80 a | 10.42 | 50.19 b | 13.76 | 62.14 c | 13.59 | 64.73 c | 15.20 |
| Mn (ppm) | 208.3 b | 144.8 | 126.0 a | 122.1 | 192.5 a | 133.5 | 253.4 c | 127.7 |
| Zn (ppm) | 30.70 a | 10.97 | 38.67 b | 13.39 | 38.16 b | 11.26 | 36.46 b | 11.57 |

The means with different letters in the same row indicate significant differences among the different phenological stages ($p < 0.05$).

When comparing the different phenological stages, the Ca, Mg, B, Fe, Na and Cl concentrations rose, and N, P, K, S, and Cu lowered as the study period advanced. These results are similar to other previously reported leaf nutrient concentrations for persimmon fruit [41,42]. Mn was the only element to decrease and then increase again. Clark and Smith [41] also observed this behaviour in 'Fuyu' persimmon. Similar to other fruit trees

grown in calcareous soil, persimmon is sensitive to Mn and Zn deficiency. The correction of these compounds is usually made by foliar applications to cover needs [7]. Therefore, Mn and Zn showed no clear increasing or decreasing trend because the foliar applications of these nutrients were not applied homogeneously in all the plots. In addition, Mn is one of the active principles of Mancozeb [43], the main fungicide used in the Valencian Community for controlling circular leaf spots (CLS) in persimmon caused by *Mycosphaerella nawae* Hiura & Ikata. Since August 2008, CLS has become the most important persimmon foliar disease in Spain and leads to severe yield losses upon harvest [44]. This fungicide is applied mainly in spring when *M.Nawae* ascospores are released from leaf litter [45].

Table 3 shows significant differences in the concentration of most nutrients among the phenological stages during the study period. A bigger difference was obtained between the mean N and K values, with significant differences noted in each phenological state. Nutrients Ca, Mg, Cu, Fe and Zn, and those related to salinity (Na and Cl), remained constant from the FC stage.

The NOR in other crops has been previously established by the DRIS method taking leaves from a specific phenological stage. Guimaraes et al. [46] sampled ratoon sugarcane at harvest, and Serra et al. [16] also did during cotton crop flowering. Nevertheless, these results reveal that establishing a single NOR for the whole vegetative cycle will lead to wide ranges and, therefore, these ranges will not often cover the specificity of each phenological stage. Thus, the differences observed among different phenological stages reveal the need to establish distinct NOR reference tables for each studied phenological time (Tables 4–7). The main optimum ranges of leaves at FE coincide with the nutrient concentration observed by Pomares et al. [7] in July, which is when the FE of the PDO 'Kaki Ribera del Xúquer' takes place. Nevertheless, the NORs for FE and HV were slightly lower than the ranges reported by George et al. [21], whose results might be lower because they employed a different cultivar, cv 'Fuyu' compared to 'Rojo Brillante', and edaphic conditions.

**Table 4.** Nutrient optimal range (NOR) in the AF stage of persimmon fruit in PDO 'Kaki Ribera del Xúquer', Spain.

| | **After Flowering (AF)** | | | | |
|---|---|---|---|---|---|
| | **Deficient** | **Low** | **Optimum** | **High** | **Excessive** |
| N (%) | ≤1.878 | 1.879–1.989 | 1.990–2.211 | 2.212–2.322 | ≥2.323 |
| P (%) | ≤0.120 | 0.121–0.127 | 0.128–0.143 | 0.144–0.150 | ≥0.151 |
| K (%) | ≤1.263 | 1.264–1.442 | 1.443–1.802 | 1.803–1.982 | ≥1.983 |
| Ca (%) | ≤0.386 | 0.387–0.570 | 0.571–0.939 | 0.940–1.22 | ≥1.123 |
| Mg (%) | ≤0.200 | 0.201–0.253 | 0.254–0.361 | 0.362–0.415 | ≥0.416 |
| Na (%) | ≤0.007 | 0.008–0.009 | 0.010–0.013 | 0.014–0.015 | ≥0.016 |
| S (%) | ≤0.177 | 0.178–0.200 | 0.201–0.247 | 0.248–0.269 | ≥0.270 |
| Cl (%) | ≤0.491 | 0.492–0.203 | 0.204–0.396 | 0.397–0.492 | ≥0.493 |
| B (ppm) | ≤13.07 | 13.08–26.43 | 26.44–53.16 | 53.17–66.52 | ≥66.53 |
| Cu (ppm) | ≤2.39 | 2.40–3.96 | 3.97–7.10 | 7.11–8.66 | ≥8.67 |
| Fe (ppm) | ≤28.90 | 28.91–35.85 | 35.86–49.75 | 49.76–56.70 | ≥56.71 |
| Mn (ppm) | ≤15.19 | 15.20–111.75 | 111.76–304.89 | 304.90–401.45 | ≥401.46 |
| Zn (ppm) | ≤16.07 | 16.08–23.38 | 23.39–38.01 | 38.02–45.32 | ≥44.33 |

**Table 5.** Nutrient optimal range (NOR) in the FE stage of persimmon fruit in PDO 'Kaki Ribera del Xúquer', Spain.

| | Deficient | Low | Optimum | High | Excessive |
|---|---|---|---|---|---|
| **Fruit Enlargement (FE)** | | | | | |
| N (%) | ≤1.723 | 1.724–1.826 | 1.827–2.033 | 2.034–2.136 | ≥2.137 |
| P (%) | ≤0.094 | 0.095–0.099 | 0.100–0.109 | 0.110–0.114 | ≥0.115 |
| K (%) | ≤1.198 | 1.199–1.318 | 1.319–1.559 | 1.560–1.679 | ≥1.680 |
| Ca (%) | ≤1.437 | 1.438–1.595 | 1.596–1.911 | 1.912–2.069 | ≥2.070 |
| Mg (%) | ≤0.418 | 0.419–0.480 | 0.481–0.604 | 0.605–0.666 | ≥0.667 |
| Na (%) | ≤0.008 | 0.009–0.010 | 0.011–0.017 | 0.018–0.020 | ≥0.021 |
| S (%) | ≤0.187 | 0.188–0.214 | 0.215–0.269 | 0.270–0.296 | ≥0.297 |
| Cl (%) | ≤0.543 | 0.544–0.630 | 0.631–0.805 | 0.806–0.892 | ≥0.893 |
| B (ppm) | ≤15.97 | 15.98–24.38 | 24.39–41.23 | 41.24–49.65 | ≥49.66 |
| Cu (ppm) | ≤1.65 | 1.66–2.72 | 2.73–4.86 | 4.87–5.93 | ≥5.94 |
| Fe (ppm) | ≤31.97 | 31.98–41.07 | 41.08–59.30 | 59.31–68.41 | ≥68.42 |
| Mn (ppm) | ≤0.00 | 0.00–44.67 | 44.68–207.42 | 207.43–288.79 | ≥288.80 |
| Zn (ppm) | ≤20.81 | 20.82–29.73 | 29.74–47.59 | 47.60–56.52 | ≥56.53 |

**Table 6.** Nutrient optimal range (NOR) in the FC of persimmon fruit in PDO 'Kaki Ribera del Xúquer', Spain.

| | Deficient | Low | Optimum | High | Excessive |
|---|---|---|---|---|---|
| **Fruit Colouring (FC)** | | | | | |
| N (%) | ≤1.447 | 1.448–1.571 | 1.572–1.821 | 1.822–1.945 | ≥1.946 |
| P (%) | ≤0.080 | 0.081–0.090 | 0.091–0.110 | 0.111–0.119 | ≥0.120 |
| K (%) | ≤0.917 | 0.918–1.147 | 1.148–1.607 | 1.608–1.837 | ≥1.838 |
| Ca (%) | ≤1.811 | 1.812–2.214 | 2.215–3.020 | 3.021–3.423 | ≥3.424 |
| Mg (%) | ≤0.441 | 0.442–0.550 | 0.551–0.770 | 0.771–0.879 | ≥0.880 |
| Na (%) | ≤0.010 | 0.0114–0.013 | 0.014–0.021 | 0.022–0.024 | ≥0.025 |
| S (%) | ≤0.154 | 0.155–0.184 | 0.185–0.244 | 0.245–0.274 | ≥0.275 |
| Cl (%) | ≤1.001 | 1.002–1.121 | 1.122–1.362 | 1.363–1.483 | ≥1.484 |
| B (ppm) | ≤33.41 | 33.42–45.70 | 45.71–70.30 | 70.31–82.59 | ≥82.60 |
| Cu (ppm) | ≤1.56 | 1.57–2.29 | 2.30–3.77 | 3.78–4.51 | ≥4.52 |
| Fe (ppm) | ≤44.02 | 44.03–53.08 | 53.09–71.21 | 71.22–80.27 | ≥80.28 |
| Mn (ppm) | ≤14.46 | 14.47–103.46 | 103.47–281.47 | 281.8–370.47 | ≥370.48 |
| Zn (ppm) | ≤23.41 | 23.15–30.65 | 30.66–45.67 | 45.68–53.17 | ≥53.18 |

**Table 7.** Nutrient optimal range (NOR) at HV stage of persimmon fruit in PDO 'Kaki Ribera del Xúquer', Spain.

| | Deficient | Low | Optimum | High | Excessive |
|---|---|---|---|---|---|
| **Harvest (HV)** | | | | | |
| N (%) | ≤1.256 | 1.257–1.338 | 1.339–1.503 | 1.504–1.585 | ≥1.586 |
| P (%) | ≤0.073 | 0.074–0.078 | 0.079–0.089 | 0.090–0.094 | ≥0.095 |
| K (%) | ≤0.737 | 0.738–0.904 | 0.905–1.240 | 1.241–1.407 | ≥1.408 |
| Ca (%) | ≤2.111 | 2.112–2.278 | 2.279–2.614 | 2.615–2.781 | ≥2.782 |
| Mg (%) | ≤0.508 | 0.509–0.578 | 0.579–0.720 | 0.721–0.790 | ≥0.791 |
| Na (%) | ≤0.010 | 0.011–0.012 | 0.013–0.017 | 0.018–0.019 | ≥0.020 |
| S (%) | ≤0.149 | 0.150–0.169 | 0.170–0.211 | 0.212–0.231 | ≥0.232 |
| Cl (%) | ≤0.992 | 0.993–1.095 | 1.096–1.302 | 1.303–1.405 | ≥1.406 |
| B (ppm) | ≤25.61 | 25.62–47.43 | 47.44–91.09 | 91.10–112.92 | ≥113.93 |
| Cu (ppm) | ≤1.53 | 1.54–2.52 | 2.56–4.52 | 4.53–5.51 | ≥5.52 |
| Fe (ppm) | ≤44.46 | 44.47–54.59 | 54.60–74.87 | 74.88–85.00 | ≥85.01 |
| Mn (ppm) | ≤83.20 | 83.21–168.31 | 168.32–338.54 | 338.55–423.66 | ≥423.67 |
| Zn (ppm) | ≤21.03 | 21.04–28.74 | 28.75–44.17 | 44.18–51.89 | ≥51.90 |

### 3.2. Selecting the Sampling Period

In order to determine the optimum sampling time to know the more balanced nutritional period of persimmon plants, production was taken into account by separating data into two groups based on the total average production. With an average of 40 t·ha$^{-1}$, a block of 28 high-yielding orchards and a block of 30 low-yielding orchards were established. The leaf analysis not only depends on correct crop stage selection but also on correctly choosing leaf type or knowing the influence of cultivation conditions, such as irrigation type [47]. Data were divided according to irrigation system type, flood (FL) or drip (D), and the sprout origin, vegetative (V) or floral (F).

The NBI can be applied to identify deficiencies, excesses or nutritional balances of macronutrients (N, P, K, Ca, Mg, S) and micronutrients (Fe, Mn, Zn, Cu, Mo, B) in leaf tissue [48]. Thus, to establish the optimum sampling time, the DRIS method tool of the NBI was used. In this study, the NBI was determined in the four different phenological stages after taking into account yield, sprout type and irrigation (Table 8). The data showed that only the samples from the vegetative sprouts of the drip-irrigated orchards with yields over 40 t·ha$^{-1}$ have no significant differences among phenological stages. NBIs were lower in all cases during FE, with significant differences compared to the other phenological groups. The second most balanced phenological stage was FC, which had no significant differences with FE, except for the samples from the flood-irrigated orchards from vegetative sprouts with yields above 40 t·ha$^{-1}$. The most unbalanced stage was AF. As AF was sampled in May, the highest NBI was obtained because the period from March to June is characterized by the high nutrient requirements needed for sprouting, flowering and fruit set [7].

**Table 8.** Nutrient Balance Index (NBI) in the different phenological stages of 'Rojo Brillante' persimmon.

|  |  | AF | FE | FC | HV |
|---|---|---|---|---|---|
| <40 t·ha$^{-1}$ | FL × V | 15.66 d | 4.29 a | 6.34 b | 9.84 c |
|  | FL × F | 12.54 c | 4.55 a | 6.12 a | 8.88 b |
|  | D × V | 15.26 c | 4.39 a | 6.99 ab | 9.61 b |
|  | D × F | 10.98 b | 4.64 a | 6.77 ab | 8.48 ab |
| >40 t·ha$^{-1}$ | FL × V | 15.24 c | 5.94 a | 8.54 ab | 11.09 b |
|  | FL × F | 12.06 b | 5.59 a | 7.54 a | 8.04 a |
|  | D × V | 11.76 a | 6.57 a | 9.84 a | 8.79 a |
|  | D × F | 7.85 ab | 5.48 a | 5.61 a | 9.60 b |
| A: Irrigation |  | ns | ns | ns | ns |
| B:Sprout |  | * | ns | ns | ns |
| C: Yield |  | ns | * | ns | ns |
| AB |  | ns | ns | ns | ns |
| AB |  | ns | ns | ns | ns |
| BC |  | ns | ns | ns | ns |
| ABC |  | ns | ns | ns | ns |

The means with different letters in the same row indicate significant differences among the different phenological stages [AF, FE, FC and HV] due to the evaluated factors [(irrigation system (A), flush type (B) and yield (C)].ns denotes there is not significant differences and * indicates significant differences at $p < 0.05$.

Regarding the subgroups into which data were divided, the sprout location of samples was a determinant in the AF stage. The leaves from vegetative sprouts were more unbalanced. In all cases, the content of nutrients in the leaves from fruiting shoots was lower (data not shown). This could be because fruiting shoots have started fruit set, which induces leaves to change from being a sink to a source of nutrients and photosynthates [49]. Nevertheless, sprout types showed no significant differences in the other phenological stages. This agrees with Clark and Smith [42] because these authors did not observe any major differences between the nutrient concentrations measured in the leaves sampled from fruiting or non-fruiting shoots. This finding suggests that the leaves in close proximity to fruit are not subjected to disproportionate nutrient stress during fruit development.

Approximately in the FE stage, and despite NBIs being low in all cases, the yield had a significant effect. Upon sampling (July), the high-yielding persimmon trees (>40 t·ha$^{-1}$) had greater nutrient mobility than the low-yielding trees. These trees had higher nutrient requirements because they had a larger fruit set.

To know the order of the most unbalanced nutrients in each phenological stage, and bearing in mind that sprout was a determinant of AF and yield in FE, Figure 5 shows the mean DRIS indices of the total population for each nutritional status. Nutrient excess or deficiency varies in order and severity terms depending on sampling time. As observed in the different NBIs, the greatest nutrient excesses and deficiencies are for AF and HV.

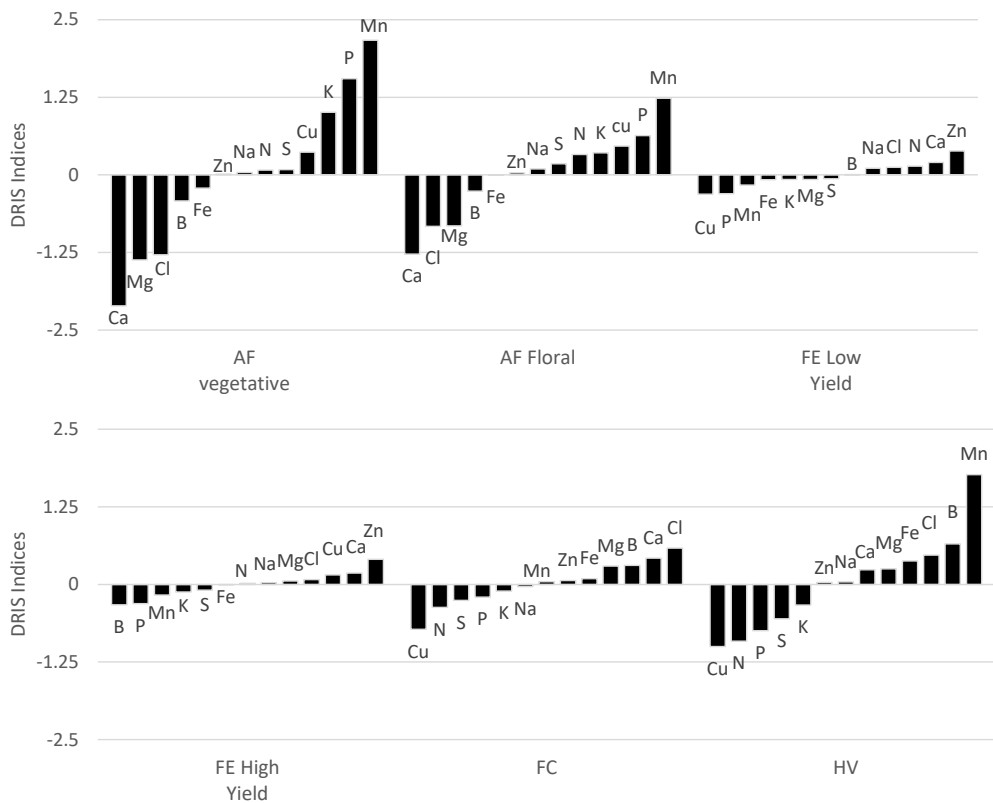

**Figure 5.** Limiting order of nutrients of the total population in each phenological stage (AF; FE; FC; HV) according to the DRIS indices determined using the whole database (high- and low-yielding populations) of 'Rojo Brillante' persimmon PDO 'Kaki Ribera del Xúquer', Spain.

At the AF sampling time, and similar to the results in Table 8, the leaves from vegetative sprouts exhibited greater nutrient imbalances than those from floral sprouts. Nevertheless, in both cases, the lowest indices were for Ca, Cl and Mg. This does not imply that they are limiting for crops; rather, leaves did not have enough time to uptake these nutrients.

Of these elements, Ca is a highly immobile element [50,51]. Mg is the central atom in the chlorophyll molecule that is needed for its biosynthesis. It increases throughout the vegetative cycle because it is known that chlorophyll content in young leaves is lower than in old leaves [52]. Regarding Cl, although its deficiency is presented, it appears in excess in the other phenological stages, which verifies this crop's salinity sensitivity.

Conversely, a larger quantity of Mn appears in AF and suggests it is excessive. Despite this nutrient occupying intermediate positions at FE and FC, Mn once again appeared in excessive amounts at HV. As indicated above, this is because persimmon trees are foliar-sprayed with Mn at different vegetative cycle times. With the obtained results, it can be established that the first Mn application is necessary because the Mn level is not excessive as harvest advances. However, an attempt should be made to reduce the last treatment as much as possible. The growing cycle ends with an excessive Mn concentration and,

as persimmon is a deciduous crop, leaves fall off when the cycle ends, and part of the immobile elements present in leaves are nutrients that return to the soil. It could induce excessive Mn concentrations in new plant tissue that can alter several processes, such as enzyme activity, absorption, translocation and utilisation of other mineral elements (Ca, Mg, Fe and P), which cause oxidative stress [53,54].

During FE, apart from the differences observed between the NBIs of the high-yielding population vs. the low-yielding population, Figure 5 shows that the order of deficiency or excess is presented by $I_{DRIS}$ changes. While Cu was deficient only in the low-yielding orchards, B was deficient only in the high-yielding orchards. Both these elements could be determinants for knowing final 'Rojo Brillante' persimmon yields. S, P and K appeared as limiting deficiency elements in both populations. Of these, P was limited from FE and S in all the studied nutritional stages. In the last two phenological stages, nutrients were unbalanced and not limiting for the period from July to September, as this period requires considerable nutrients to cover full fruit development. Because of this, these nutrients were already used by fruit sets. Nevertheless, when paying attention to K, and despite persimmon crops usually being fertilised with this nutrient during FE, plants presented K deficiency. Thus, K fertilisation could be advisable to improve persimmon production. However, K fertilisation must be cautiously carried out because it can decrease Ca uptake and subsequently affect fruit firmness or fruit conservation [7].

Both P and S are limiting nutrients from FE, which is the more balanced sampling period, until the end of the study period. Increasing P deficiency from flowering to harvest has also been observed by Gosavi et al. [47], who attributed it to not only developing fruit (cell division), demanding more P, but also to limiting P supply in soil because of high pH and calcareousness.

## 4. Conclusions

DRIS norms were established for 'Rojo Brillante' persimmon PDO 'Kaki Ribera del Xúquer' (Spain). For each pair of nutrients, a single norm was selected for the whole vegetative period from flowering to fruit harvest.

The correlation established between the DRIS indices obtained with DRIS norms and the concentrations of nutrients in the leaves of the high-yielding population verified that a single norm for each nutrient ratio is correct.

The different employed regression equations showed very high coefficients of determination ($R^2$), which indicates that the evolution of the relationship between nutrients is not linear in most cases because it depends on the evolution of different nutritional elements with one another.

Although there were many overlaps in the range of optimal values for each phenological state, they did not completely coincide in any case. Therefore, establishing tables with the NOR for each phenological state is considered necessary. These tables will be useful for avoiding wide ranges that hide certain deficiencies or excesses or ranges that would be obtained if the complete vegetative cycle were considered. In addition, based on the DRIS method, reference tables with the NOR values for the cultivation of 'Rojo Brillante' persimmon in the Mediterranean area has been established for the first time, which will be used for the practical cultivation of this variety.

Trees show a totally different NBI depending on the sampling time. The phenological stage when the 'Rojo Brillante' persimmon trees are more balanced is FE (July). However, the influence of yield is shown during this period. The greater imbalance in the trees with a higher yield indicates that they require greater nutrient mobility from leaves to fruits.

Leaf sample origin (vegetative/reproductive branch; flood-/drip-irrigated field) does not influence NBIs, except for the leaves sampled after flowering. Therefore, since FE, both sprouts can be sampled without having to differentiate DRIS standards.

The application of the results obtained for the nutritional diagnosis of the cultivar 'Rojo Brillante' should be validated in other persimmon cultivars and growing conditions.

This study establishes the basis for the nutritional diagnosis of 'Rojo Brillante' persimmon, which will allow optimal fertilisation recommendations to be made in order to achieve balanced plantations with higher production. All of this contributes to achieving the objectives pursued by international legislation in terms of balanced fertiliser application and sustainable nutrient management, as well as better management of nitrogen and phosphorus throughout their life cycle.

**Author Contributions:** Conceptualization, A.Q., B.M.-A. and J.M.; methodology, A.Q., J.M. and R.C.; formal analysis, A.Q., J.M., I.R.-C. and B.M.-A.; investigation A.Q., J.M., I.R.-C., B.M.-A. and R.C.; resources, A.Q. and R.C.; data curation, J.M. and I.R.-C.; writing—original draft preparation, J.M. and A.Q.; writing—review and editing, J.M., B.M.-A., R.C. and A.Q.; funding acquisition, A.Q. All authors have read and agreed to the published version of the manuscript.

**Funding:** This research was co-funded by Cooperativa Agrícola Ntra. Sra. del Oreto Coop. V (CANSO) in the framework of the IVIA-CANSO collaboration through Project GVA-IVIA nº 5940/7008.

**Data Availability Statement:** Not applicable.

**Acknowledgments:** We would like to thank to Mario Vendrell, Quality Chief at Cooperativa Agrícola Ntra. Sra. del Oreto Coop. V.; to Ivan Ballester and Francisco Motilla Ferrer (professional farmers) to vegetable material. Particular thanks to Enric Alcayde from Technology Transfer Service of Valencian Government for their invaluable field assistance.

**Conflicts of Interest:** The authors declare no conflict of interest. The funders had no role in the design of the study; in the collection, analyses, or interpretation of data; in the writing of the manuscript, or in the decision to publish the results.

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
