# Peer review of "DRIS Norms and Sufficiency Ranges for Persimmon ‘Rojo Brillante’ Grown under Mediterranean Conditions in Spain"

_agronomy, doi:10.3390/agronomy12061269_

Round 1
Reviewer 1 Report
This MS shows the effectiveness of DRIS norms for 'Rojo Brillante' , and addresses an important technical issue. But it needs a revision to be acceptable.
L201 The reason making the '46.38 tha-1 ' the criterion of high-yielding might not be clear. In addition, the averages shown in Table 1 might be less meaningful. The basis of high-yielding must be cleared.
Table 1 The numbers from ’Mg/N’ to 'P/Cl' should be corrected.
L214 Whether the ratio is high or low is not meaningless.
L434 Authors should mention concretely how the DRIS norms are used for the practical cultivation of 'Rojo' Brillante.'
And in addition, they should mention whether the DRIS norms in this study could be used for other persimmon cultivars/varieties.
Author Response
Dear Reviwer;
The replies are shown below in the body of the message.
This MS shows the effectiveness of DRIS norms for 'Rojo Brillante' , and addresses an important technical issue. But it needs a revision to be acceptable.
L201 The reason making the '46.38 tha-1 ' the criterion of high-yielding might not be clear. In addition, the averages shown in Table 1 might be less meaningful. The basis of high-yielding must be cleared.
The explanations has been exposed in the matherial and method sections (Lines 161-163).
Regarding table 1, the same number of decimals in the averages has been maintained, since at least a third decimal number has beennecessary for all ratios to have at least one significant value.
Table 1 The numbers from ’Mg/N’ to 'P/Cl' should be corrected.
According to the comment, it has been modified in the corrected manuscript.
L214 Whether the ratio is high or low is not meaningless.
We appreciate your comment. It has been removed from the manuscript
L434 Authors should mention concretely how the DRIS norms are used for the practical cultivation of 'Rojo' Brillante.' And in addition, they should mention whether the DRIS norms in this study could be used for other persimmon cultivars/varieties.
To emphasise this comment, the conclusions have been modified to highlight the practical use to optimise fertilisation management of 'Rojo Brillante' persimmon. However, studies would be necessary to validate its use in other cultivars and agro-climatic conditions, as has also been pointed out in the conclusion.
Reviewer 2 Report
The innovation of this paper is general, but it has not been reported in persimmon research, which has certain production guidance and practical significance.
There are some mistakes in the writing format of the paper. Please modify it carefully. For example, Diospyros kaki should be italicized.
Author Response
Dear Reviwer;
Following your kind corrections, the manuscript has been carefully revised. In addition, as it is indicated that the introduction could be improved, it has been modified for better understanding.
Reviewer 3 Report
This manuscript could be suitable for the publication in Agronomy Journal DRIS Norms and Sufficiency Ranges for Persimmon ‘Rojo Brillante’
Grown Under Mediterranean Conditions in Spain, however, the authors are requested to address the following comments while revising the manuscript. The study is within the scope of the journal. The study shows deficiencies and would be significantly improved with the addition of more details about:
- hypothesis in the manuscript
- research questions (RQs) in the end introduction section;
- fertilizers dose ranges should be included in the pure ingredient, kgꞏha-1
In the method section, please provide the procedure for correlation and regression analysis stating the platform where the analysis has been done.
I strongly recommend a separate discussion section. In the section, the authors need to discuss their key findings in the context of existing literature. A separate discussion section to make the paper a stronger article.
Conclusions
The conclusion should be presented in line with RQs.
Author Response
Dear editor;
Thank you for your corrections and feedback. The manuscript has been carefully revised and the replies are shown in the body of the message.
Hypothesis in the manuscript and research questions (RQs) in the end introduction section;
- The main hypothesis of this study is the design of nutritional diagnosis tools for persimmon cv. 'Rojo Brillante' based on the DRIS methodology that allow a more sustainable fertilisation management in line with the demands of international legislation, and more specifically the EU in its 2030 agenda. To highlight this hypothesis, as well as the RQs, the introduction has been modified.
Fertilizers dose ranges should be included in the pure ingredient, kgꞏha-1
- Commonly used units for establishing fertiliser rates are: N, P2O5, K20, MgO and Fe. They are not usually used in reference to individual elements.
In the method section, please provide the procedure for correlation and regression analysis stating the platform where the analysis has been done.
- In line with your comment, the description of the statistical analysis has been modified for better understanding.
I strongly recommend a separate discussion section. In the section, the authors need to discuss their key findings in the context of existing literature. A separate discussion section to make the paper a stronger article.
- This manuscript, although relevant, is long and complex to read. Therefore, to present the results and discussion in two separate sections would require repeating the explanation of the results in the discussion, which would make the manuscript longer and complicate its comprehension.
Conclusions
The conclusion should be presented in line with RQs.
- In agreement with your suggestion on improving both the study hypothesis and the RQs in the introduction, we have been able to improve the final conclusion in line with the RQs.
Round 2
Reviewer 2 Report
Accept in present form
Reviewer 3 Report
Accept in present form.